# Germination and Growth of Plasma-Treated Maize Seeds Planted in Fields and Exposed to Realistic Environmental Conditions

**DOI:** 10.3390/ijms24076868

**Published:** 2023-04-06

**Authors:** Nina Recek, Rok Zaplotnik, Alenka Vesel, Gregor Primc, Peter Gselman, Miran Mozetič, Matej Holc

**Affiliations:** 1Jožef Stefan Institute, Jamova Cesta 39, 1000 Ljubljana, Slovenia; 2Interkorn Ltd., Gančani 94, 9231 Beltinci, Slovenia

**Keywords:** maize, plasma, yield, seedling emergence, field, fungicide, eco-layer, realistic field conditions

## Abstract

In this study, we applied an inductively coupled, radio frequency oxygen plasma to maize seeds and investigated its effects on seedling emergence, plant number at tasseling, and crop yield of maize in realistic field conditions. Maize seeds of seven different hybrids were treated over two harvest years. In addition to plasma-treated seeds, a control sample, fungicide-treated seeds, an eco-layer, and a plasma and eco-layer combination, were planted. Seedling emergence, plant number at tasseling (plants/m^2^), and yield (kg/ha), were recorded. In the first harvest year, results were negatively affected by the presence of an insect pest. In the second harvest year, plant number and yield results were more uniform. In both years, for two and three hybrids, respectively, the highest yield arose from plants from plasma-treated seeds, but the differences were only partially significant. Considering our results, plasma treatment of maize seeds appears to have a positive effect on the yield of the plant.

## 1. Introduction

Maize, also called corn, is a cereal crop widely used throughout the world for human and animal consumption, and the production of ethanol and other corn products. Maize is the second most produced crop in the world after sugar cane; with a yearly yield of over 1100 million tonnes, it amounted to 12% of the total world production of crops in the year 2020. Currently, 50% of the world’s total yield of maize is produced in North and South America [1,2].

As with other crops, efforts are underway to stimulate maize yield in the face of uncertain climate conditions [3]. A remarkable approach, popular in the last decade, is the use of non-equilibrium gaseous plasma to treat crop seeds before planting to improve an array of germination and growth characteristics and ultimately increase yield [4]. Maize has been a popular subject of plasma agriculture publications, which have examined various aspects of plasma application to crop seeds, including decontamination of fungi, germination and growth improvement, and effects on various biochemical parameters [5,6,7,8,9]. As with other agricultural seeds, various authors have explained the potential benefits of these findings in agricultural settings. Some publications have theorized that particular changes to biochemical parameters, such as the recorded modifications of H_2_O_2_ and NO signalling in roots, have the potential for yield improvement [8]. Our hypothesis was, however, that plasma treatment would enhance the yield and germination of maize seeds. Germination would be improved due to increased water uptake as a result of a more hydrophilic surface after plasma treatment. Furthermore, due to plasma decontamination of fungi on the seed surface, these seeds would be more resistant to weather conditions and insect pests, thus, resulting in higher maize yield. Not many publications have directly addressed the possibility of plasma seed treatment to improve maize yield. Collected in Table 1 are the only three publications that have thus far investigated the yield of maize after plasma treatment of its seeds.

Ahn, Gill, and Ruzic [10] have focused on the yield of maize grown in field conditions. They applied three different plasma treatments to the seeds: a dielectric barrier discharge (DBD) using helium, a microwave (MW) discharge using helium/nitrogen, and a low-pressure RF discharge in nitrogen. Differences among the yield results, recorded in bushels per hectare, were ascribed to treatment specifics such as temperature. However, none of the treatments was able to increase the yield compared to the control sample of untreated seeds. The authors attributed this result to the nearly 100% germination rate of the selected maize hybrid, the variation of environmental effects in real planting conditions, as well as the use of a protective insecticide/biological coating.

Filatova et al. [5] have worked with low-pressure RF discharge in ambient air. They recorded 1000-grain weight as a yield marker and achieved a modest increase in that parameter. In addition to an increase in the germination rate and some biochemical parameters, they have shown that plasma treatment can decrease fungal infection on seed, in this case, the presence of artificially inoculated *Penicillium* and *Fusarium* spp., as well as reduce the fungal disease infection level of plants in the field.

Finally, Karmakar et al. [9] have investigated a number of effects on the maize grain and plant after treatment with a low-frequency glow discharge (LFGD) using argon and oxygen. As a measure of yield, the number and weight of grains per cob were recorded, and both were notably improved in maize grown from plasma-treated seeds. In addition, the germination rate, a number of growth parameters, as well as several biochemical parameters, were improved and the activity of antioxidant enzymes in leaves and roots was increased. The nutritional value of maize grown from plasma-treated seeds remained unchanged.

In our experiments, we worked with a total of seven hybrids of maize. Regarding yield parameters, we recorded plant number (no. of plants per m^2^) at emergence and at tasseling time, as well as yield (kg/ha). Maize grown from untreated seeds served as the control treatment, while seeds with other treatments were also used. The seeds were treated in an industrial-size plasma reactor which allowed the treatment of 1 kg of maize seeds per batch.

## 2. Results

### 2.1. Number of Plants

The results of the seedling emergence of the individual hybrids in the harvest year 2020 are given in Figure 1a. Results indicate that seedling emergence was poor despite high sowing density (8.5 kernels/m^2^). The best treatment condition regarding the number of emerged seedlings was the application of fungicide, except for the hybrid P9537, where the highest number of emerged seedlings was seen for seeds pre-treated with oxygen plasma followed by the application of an eco-layer. As seen in Figure 1a, some of the differences are statistically significant, especially for hybrids P9241 and P9757. The results for P9537 show no significant difference, while for P9234, only the value for fungicide treatment is statistically higher than the remaining values.

An even lower number of plants was counted at tasseling time, as seen in Figure 1b. There, the differences according to the treatment conditions seen in Figure 1a no longer apply, and there is no common trend regarding treatments seen for the different hybrids. While there are statistically significant differences in the results of hybrids P9241 and P9234, this is not the case for hybrids P9537 and P9757.

The correlation between the number of emerged maize seedlings and the number of maize plants at tasseling is shown in Figure 2. In all correlation plots hereafter, each hybrid is presented with a different color. All treatments, including the control, are pooled into each hybrid; therefore, there are 5 points (4 treatments + 1 control) of each color. We see that there is some correlation but the results are quite scattered. A higher number of seedlings at emergence does not necessarily indicate a higher number of plants at tasseling. Furthermore, from Figure 1a we cannot conclude that there is any combination effect with microorganisms and insects.

The reason for low plant count in this harvest year is due to the presence of *Agriotes* spp., a soil insect pest in which maize is typically affected. The pest significantly reduced the emergence of seedlings, as well as the number of maize plants at tasseling time. Neither the fungicide nor the eco-layer, which were applied to maize seeds with the purpose of controlling soil pests, had an effect on *Agriotes* spp. The pest affected all maize hybrids, regardless of the treatment conditions. While some hybrids and treatments were affected more and some less, the damage from *Agriotes* spp. were distributed randomly, which also explains the lack of correlation in Figure 2. Plots of maize hybrid P9241, treated with plasma and eco-layer, were so heavily affected by *Agriotes* spp. that there were almost no plants left at tasseling time. Similarly, maize hybrid P9234, with only the eco-layer applied, had very few plants left on its plots at tasseling time.

Results of seedling emergence in the next harvest year, 2021, are shown in Figure 3a, while plant counts at tasseling time are shown in Figure 3b. Figure 3a shows no major differences in the number of emerged seedlings between different hybrids. There are, however, some differences between treatment groups of the same hybrid. For hybrid P9610, seeds treated with plasma and the addition of the eco-layer performed best, but the difference is not statistically significant. For hybrids P9537 and P9978, the best-performing seeds are the plasma-treated seeds, but the number of plants is only significantly higher from the eco-layer-treated seeds in the case of P9537 and does not differ significantly from any other treatment in the case of P9978. No maize hybrid performed similarly to another hybrid regarding the treatment conditions. Compared to the previous year, the number of emerged seedlings, especially the number of plants at tasseling time, is much higher. This is attributed to the fact that in this harvest year, the soil pest *Agriotes* spp. did not affect the young plants. This is also seen in Figure 3b, where the number of plants at tasseling time is comparable to the number of seedlings at the time of emergence. The differences in plant count between treatments at tasseling time are not statistically significant for any of the hybrids.

Figure 4 shows the correlation between the number of emerged maize seedlings and plants at tasseling time in the harvest year 2021. As previously observed from comparing the number of plants in Figure 3a,b, the two values correlate quite well. This confirms that no serious damage from insects or other pests was observed from emergence to tasseling. The minimal differences in correlations may be due to weather conditions, soil quality, and other uncontrollable conditions of the field experiment.

To the best of our knowledge, this is the first paper considering the emergence of maize in the field, or the number of plants at a later point of field growth, after plasma treatment of maize seeds. In fact, not many authors have directly considered the effect of plasma treatment on seedling emergence in the field. Similar to the work presented here, we recently worked with plasma-treated wheat seeds in the field but found no significant improvements in seedling emergence among several cultivars [11]. Conversely, Pérez Pizá et al. have found improved seedling establishment parameters, such as dynamics of cumulative emergence, emergence coefficient, and weighted average emergence rate, after nitrogen DBD plasma treatment of Gatton panic (*Megathyrsus maximus*) grass seeds. This indicates that, compared to control seeds, seedlings emerged from plasma-treated seeds both earlier and in greater numbers [12]. Interestingly, in a publication concerning buckwheat, plasma treatment of seeds decreased the number of emerged seedlings but at the same time greatly improved plant growth and yield [13].

Several authors have determined plasma-mediated improvement of the maize germination rate in laboratory conditions [5,6,8,9]. If conducted according to the International Seed Testing Association (ISTA) standards, results of laboratory germination tests typically correlate well with field emergence [14]. Thus, germination rate improvement may be an indicator of positive effects on maize emergence in the field. However, it should be noted that plasma treatment may also leave the germination rate unchanged and decrease it, especially with prolonged exposure [9,15].

### 2.2. Yield

Figure 5a shows maize yield in the 2020 harvest year. Hybrids P9537 and P9757 performed much better regarding yield compared to the remaining two hybrids, P9241 and P9234. The highest yield of maize of the best-performing hybrids was seen for the seeds treated with oxygen plasma, as well as seeds treated with plasma and eco-layer for hybrid P9537. Overall, the yield of all hybrids and treatments was very low due to the attack by the *Agriotes* spp. pest on young plants, as mentioned above.

As seen in Figure 5b, there is no correlation between the number of plants at tasseling time and the yield of maize. While this lack of correlation may seem unexpected, we must consider that maize yield does not depend solely on the number of plants in the field but also on yield components, such as the number of cobs per plant, kernel rows per cob, kernels per row, and kernel weight [16]. In this harvest year, some maize plants had a much higher number of cobs than others, which indicates that a lower number of maize plants can result in a higher yield and vice versa. An example of this is hybrid P9234, where the eco-layer treatment resulted in a very low number of plants at tasseling time, while there was no significant drop in the yield. A similar finding was made by Ivankov et al. for buckwheat, where seedling emergence in the field was lower after treatment of seeds with a capacitively-coupled (CC) RF plasma, but several parameters of plant growth were improved, while there was also an increase in yield, namely in the number of matured seeds per plant and seed weight per plant [13]. Conversely, the control group of hybrid P9234 counted a high number of plants at tasseling time, but its yield was comparable to the yield of the eco-layer group, which had the lowest number of plants.

Maize yield in the harvest year of 2021 is shown in Figure 6a. Overall, the yield of all hybrids was higher than the previous year, with reasons being attributed to the higher number of emerged plants and number of plants at tasseling time, as well as the lack of damage to plants from soil insects or other pests. For each hybrid, some of the differences between the treatments are statistically significant.

There are no major differences in yields between the different hybrids, which is contrary to the previous year (Figure 5), where two of the hybrids exhibited much higher yields than the remaining two. In this year, the best-performing hybrid from the previous year, P9537, had a plasma-treated seed yield almost a third lower than the previous year. However, the total yield for this hybrid, measured across treatments, was nearly equal in 2020 and 2021. Interestingly, three of the hybrids, namely P9610, P9537, and P9978, show the highest yield coming from seeds treated with plasma. Nonetheless, the difference is statistically significant only for hybrid P9537. For P9610, the yield from plasma-treated seeds is comparable to the yields of seeds treated with fungicide and those treated with plasma and eco-layer. Hybrid P9978 is comparable to the yield of seeds treated with both plasma and eco-layer.

Similarly, as shown in Figure 5b, there is also no correlation between the number of plants and the yield in 2021 as presented in Figure 6b for each hybrid and treatment condition. As previously stated, maize yield does not depend solely on the number of plants in the field but also on yield components, such as the number of cobs per plant, kernel rows per cob, kernels per row, and kernel weight. Furthermore, in years 2020 and 2021, the aforementioned could be a consequence of numerous different environmental conditions, such as climate conditions, soil quality, drought, sun, etc., on account that maize grew in the outdoor field where it was not possible to control atmospheric conditions. Some of the atmospheric conditions, available on the nearest weather station, for the years 2020 and 2021 are presented in Figure 7.

No extreme temperatures occurred in 2020; contrarily, temperatures were as expected during each of the four seasons. Additionally, an important factor concerning the number of tasseling plants, yield, etc., was due to the higher than usual amount of precipitation during the summer season of 2020. In comparison, the year 2021 exhibited normal temperatures and despite very low precipitation in June, the overall average precipitation of the summer season was normal.

Previous publications regarding maize yield following seed plasma treatment do not report unified results. Karmakar et al. observed a remarkable increase of both the number and weight of grains per cob [9], while Filatova et al. achieved a slight increase in 1000-grain weight [5]. It is worth noting, both also recorded increases in the germination rate in the laboratory setting. Conversely, Ahn, Gill, and Ruzic, noted no changes or slight decrease in yield recorded in bushels per acre and attributed this result to the already near 100% germination rate [10]. Each previous publication worked with a single maize hybrid and, therefore, could not compare the potential different responses of several hybrids to one plasma treatment. The relationship between germination rate and yield is worth exploring further, especially as germination rate is commonly boosted by plasma treatment of agricultural seeds.

The effects of plasma are known to be useful in controlling insect pests on seeds and grains. For example, in wheat grains, plasma treatments were found to have a direct insecticidal effect on red flour beetles (*Tribolium castaneum*) [17], but also were shown to protect wheat seeds against the same insect during long-term storage [18]. It is, therefore, plausible that the yield increase seen in maize hybrids P9537 and P9757 was partly due to plasma-mediated protection against insect pests.

## 3. Materials and Methods

### 3.1. Seed Material

Seven different maize hybrids (P9241, P9537, P9757, P9234, P9610, P9363, and P9978) were obtained from Interkorn Ltd. (Gančani, Slovenia); the treatment of seeds was performed as described herein. After storage, the untreated and treated seeds were sown and harvested by Žipo Lenart Ltd. (Lenart, Slovenia) for two consecutive years, 2020 and 2021.

### 3.2. Plasma Reactor

The experimental setup used in this study is shown in Figure 8. A glass discharge tube 20 cm in diameter and 2 m in length was pumped using the two-stage rotary vacuum pump Trivac D 65 B (Leybold GmbH, Cologne, Germany), which has a nominal pumping speed of 65 m^3^/h. Gas was introduced through the mass flow controller Aera FC-7700 (Advanced Energy, Denver, CO, USA) on the opposite side of the discharge tube from the pump. Oxygen with 99.999% purity was used. The pressure was measured with an absolute capacitance pressure gauge MKS Baratron 722B (MKS Instruments, Andover, MA, USA). Oxygen pressure was set to 40 Pa.

Inductively coupled plasma was generated inside a 14-turn excitation coil connected to a 5 kW, 13.56 MHz RF generator Cesar 1350 (Advanced Energy, Denver, CO, USA), through an in-house made L-type matching network. In all the experiments, the forward RF power was set to 2000 W, whereas the reflected power was around 750 W. At these conditions, the plasma was in the E-mode.

To ease the manipulation of maize grains, a perforated aluminium holder was used.

### 3.3. Plasma Treatment

During each treatment, 1 kg of maize was treated simultaneously. Dry seeds were uniformly distributed over a perforated aluminium tray holder resulting in the individual seeds not contacting each other. The holder was placed in the middle of the copper coil, as shown in Figure 8. The plasma was uniformly distributed along the discharge tube; therefore, seeds were directly exposed to the glowing plasma. Treatment time was fixed to 120 s, whereas longer treatment times caused visible damage to the seeds during preliminary experiments and were thus not included in the results.

### 3.4. Eco-Coating Application

Eco-coating, consisting of the bacterium *Bacillus subtilis* (0.5% suspension) and the algae *Ascophyllum nodosum*, was prepared prior to treatment. The suspension of *Bacillus subtilis* was diluted with water in the appropriate ratio by applying 0.6 g of *Bacillus subtilis* and 0.1 mL of *Ascophyllum nodosum* per 100 g of maize seeds. Eco-layers were applied to untreated seeds in addition to plasma-treated seeds. The plasma-treated seeds had the eco-coating applied three days after plasma treatment due to the logistics between the Jožef Stefan Institute, where the plasma treatment of seeds was performed, and Interkorn Ltd., where seeds were further subjected to eco-coating. Seeds were left to dry, then packed in paper bags and sown two days after applying the eco-coating.

### 3.5. Experimental Design

Four maize hybrids (P9241, P9537, P9757, and P9234) were sown in the year 2020, and four others (P9610, P9537, P9363, and P9978) were sown in 2021. In the year 2021, the best-performing hybrid from the year 2020 was kept for re-sowing, and three new hybrids were selected and added.

A total of 20 kg of seeds of each hybrid was used in this study. Each hybrid was subjected to four treatments; thus, five groups of seeds were prepared. The first group comprised untreated control seeds; the second group was treated with the fungicide Redigo Pro (Bayer, Leverkusen, Germany); the third group was treated with plasma; the fourth group was treated with the eco-layer; and the fifth group was pre-treated with plasma and then treated with the eco-layer. All the seeds, with and without eco-layer applied, were sown two days after the last treatment.

The treatments were performed in four replications and each replication (1 kg of seeds) was sown in a separate plot. Thus, each hybrid was sown in 20 plots (5 treatments × 4 replications), totaling 80 plots per year. The plot size was 19.6 m^2^ (length 7 m and width 2.8 m) and the seeding density was 8.5 per m^2^. The field experiment was designed according to the random block system with four replications (split-plot). Seeds were sown in April. Maize seedlings were counted after emergence (end of May), and later, plants were counted at tasseling time (mid-August).

### 3.6. Emergence of Seedlings in the Field

The emergence of maize seedlings, number of seedlings per m^2^, was monitored and counted in real-life conditions in the field. No seed groups were watered before, during, or after seedling emergence, thus, all groups grew in the same conditions, depending only on the weather. The emerged seedlings were counted at the end of May. The mean number of emerged seedlings for each treated group was calculated from the number of counted seedlings on three plots of the same hybrid.

### 3.7. Maize Plants at Tasseling Time

Tasseling—a vegetative growth stage in maize development—is the emergence of a structure called the tassel, which is a branched inflorescence found at the tip of the stem [19]. The number of maize plants was counted during this growth stage in mid-August of each year. The mean number of emerged plants was calculated from the number of counted plants on three plots of the same hybrid.

### 3.8. Maize Yield after Harvest

At harvest, the yield of each replication of every maize hybrid was collected separately and expressed in kg/ha. The mean yield of each treatment of each hybrid was calculated from the yields of all four replications.

### 3.9. Statistical Analysis

Data regarding the number of plants at emergence and tasseling time, as well as yield measurements, were statistically analyzed using the JASP 0.16.3.0. open-source software (The University of Amsterdam, Amsterdam, The Netherlands). Group means were calculated and compared using ANOVA followed by the post hoc Tukey’s range test. Differences in the means were considered statistically significant at *p* < 0.05. Error bars are shown in charts and represent standard error.

## 4. Conclusions

In this study, we used a low-pressure, inductively coupled oxygen plasma, to treat maize seeds, in addition to the application of fungicide and an eco-layer. We planted the untreated and treated seeds in the field and followed their performance according to the number of plants at seedling emergence and tasseling, as well as the final yield. We used a total of seven maize hybrids: four in the first harvest year and the best-performing hybrid from the first year, and three new hybrids in the second year.

We have shown that the treatments affected maize emergence and yield in various ways. In the first harvest year, plant emergence, and consequently the yield, was affected by the presence of the insect pest *Agriotes* spp. Nonetheless, in two of the four hybrids, the statistically significant highest yield arose from plants from plasma-treated seeds. In the second harvest year, plant number was more uniform, and the variations in yield were less extreme. Again, in three of the four hybrids, the statistically significant highest yield arose from plants from plasma-treated seeds, but only one of those three differences was statistically significant.

Different maize hybrids appear to respond to the same plasma treatment differently. Another point of variation to consider is the randomly distributed effect of pest attacks and the unpredictability of field growth conditions, as evidenced by the change in plant number and yield from the harvest year 2020 to 2021. All results considered, plasma treatment of maize seeds appears to have a positive effect on the yield of the plant.

## Figures and Tables

**Figure 1 ijms-24-06868-f001:**
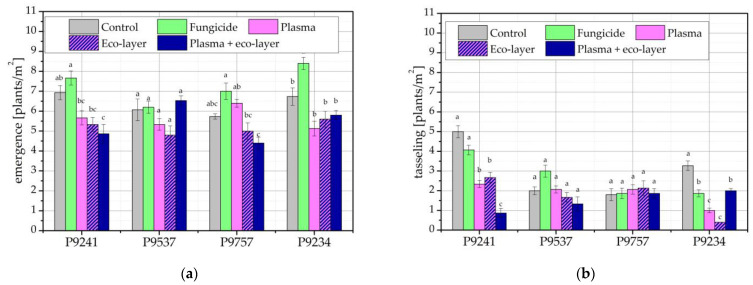
Number of maize plants in two vegetative growth stages of maize in the harvest year 2020: (**a**) at seedling emergence; and (**b**) at tasseling time. The error bars represent standard error. Different lowercase letters above data points represent statistically significant differences (*p* < 0.05; post hoc Tukey’s range test) between treatments.

**Figure 2 ijms-24-06868-f002:**
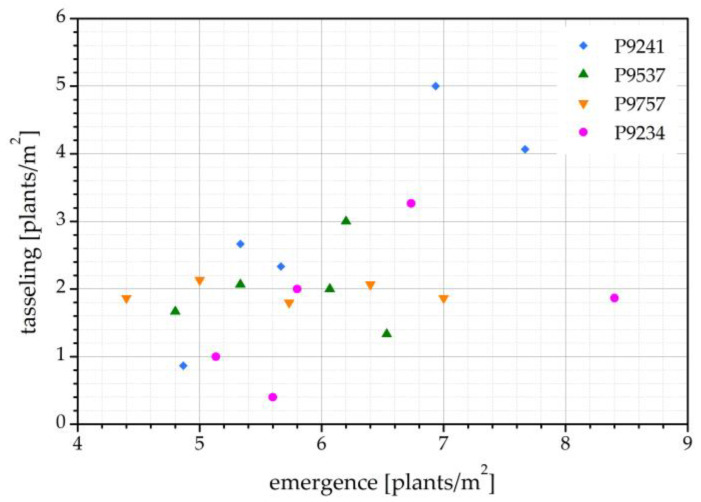
Correlation between the number of emerged seedlings (May) and the number of maize plants at tasseling (August) for the harvest year 2020.

**Figure 3 ijms-24-06868-f003:**
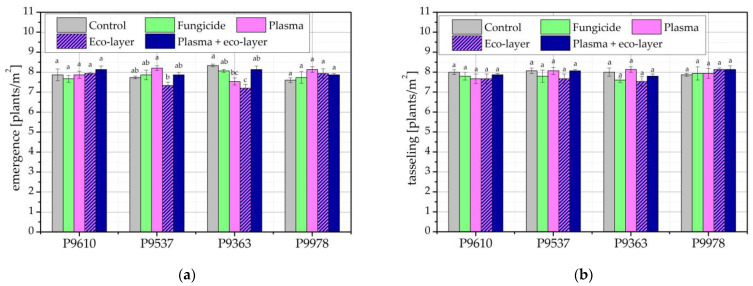
Number of maize plants in two vegetative growth stages of maize in the harvest year 2021: (**a**) at seedling emergence; and (**b**) at tasseling time. The error bars represent standard error. Different lowercase letters above data points represent statistically significant differences (*p* < 0.05; post hoc Tukey’s range test) between treatments.

**Figure 4 ijms-24-06868-f004:**
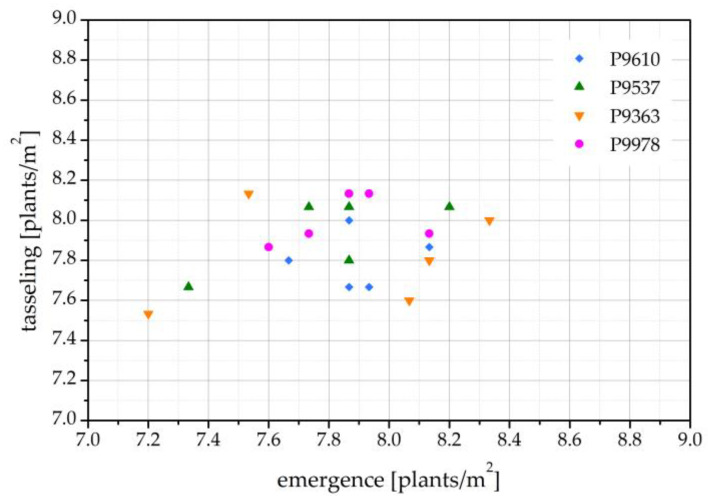
Correlation between the number of emerged seedlings (May) and the number of maize plants at tasseling (August) for the harvest year 2021.

**Figure 5 ijms-24-06868-f005:**
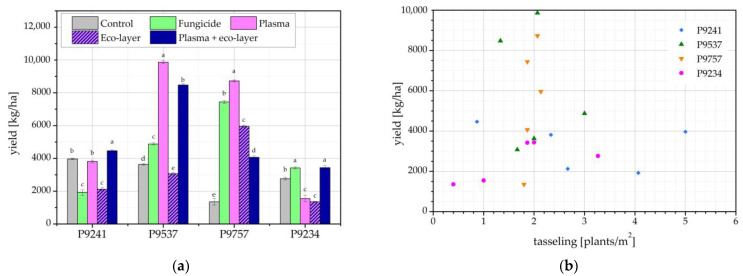
(**a**) Maize yield; and (**b**) correlation between number of plants at tasseling time and maize yield in the harvest year 2020. The error bars represent standard error. Different lowercase letters above data points represent statistically significant differences (*p* < 0.05; post hoc Tukey’s range test) between treatments.

**Figure 6 ijms-24-06868-f006:**
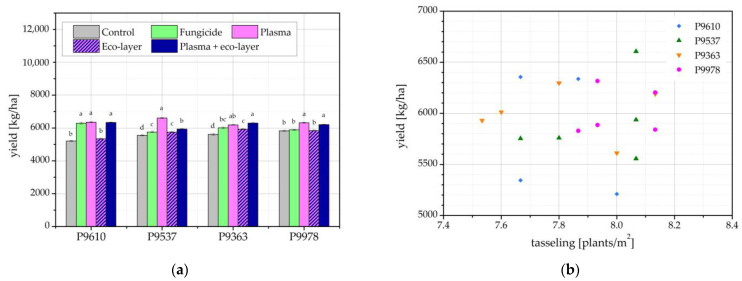
(**a**) Maize yield; and (**b**) correlation between the number of plants at tasseling time and maize yield in the harvest year 2021. The error bars represent standard error. Different lowercase letters above data points represent statistically significant differences (*p* < 0.05; post hoc Tukey’s range test) between treatments.

**Figure 7 ijms-24-06868-f007:**
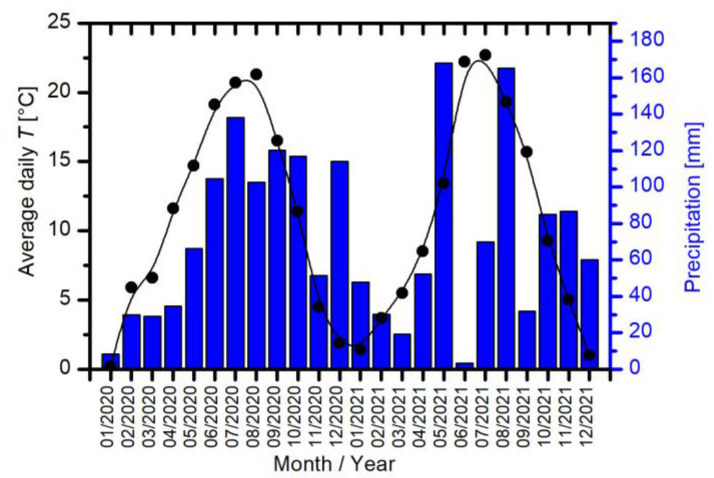
Average daily temperatures and precipitation for each month in years 2020 and 2021 measured on the nearest permanent weather station.

**Figure 8 ijms-24-06868-f008:**
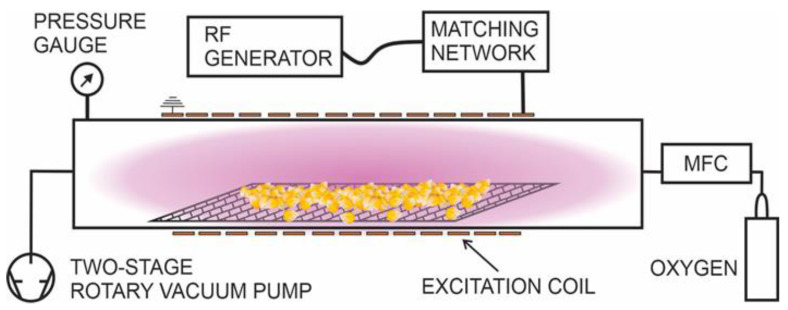
Schematic representation of industrial plasma reactor used to treat maize.

**Table 1 ijms-24-06868-t001:** An overview of publications regarding the effects of plasma treatment on maize yield. DBD: dielectric barrier discharge, MW: microwave, RF: radio frequency, LFGD: low-frequency glow discharge, GR: germination rate, S: sprout, R: root, L: leaf, G: grain, SEM: scanning electron microscopy, ↑: increase, ↓: decrease, unch.: unchanged, n/a: not available.

Author	Year	Plasma	Gas	Pressure [Pa]	Yield	↑ Yield [%]	Other Effects
Ahn [10]	2019	DBD	He	atm	unch. or ↓ bushels/acre	unch. or ↓	n/a
		MW	He/N_2_	atm			
		RF	N_2_	0.1 Torr			
Filatova [5]	2020	RF	ambient air	200 Pa	↑> 1000-grain weight	1.7%	↑> GR; ↑> S, unch./↓> R; phenols (unch. S, ↑> R), anthocyanins (↑> S, ↑> R); proline (↓> S, ↑> R), ↓> fungal infection
Karmakar [9]	2021	LFGD	Ar/O_2_	400 Torr	↑> no. of grains per cob, ↑> the weight of grains per cob	~33%, ~43%	↑> GR; ↑> S, R, fresh and dry weight, no. of leaves, stem d.; SEM: rougher surface; G: unch. nutritional value; ↑> chlorophyll conc., ↑> H_2_O_2_, NO (L, R); ↑> tot. sol. sugar, protein, phenol (L, R); ↑> Fe, Zn (L, G); ↑> antioxidant enzymes (L, R)

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
