# Peer review of "Germination and Growth of Plasma-Treated Maize Seeds Planted in Fields and Exposed to Realistic Environmental Conditions"

_ijms, 2023, doi:10.3390/ijms24076868_

Round 1
Reviewer 1 Report
Although this is a study in a realistic environment with various uncertainties, the value of this paper is in the fact that it has experimented with different types of treatments on various hybrid seeds and authors have found a certain degree of trend while being affected by pests. In this research field of plasma agriculture, there are few reports of research from this perspective, which requires a long period of time, and this research conducted over a two-year period is a suggestive contribution to this field. The disclosure of negative and positive data without selection is valuable because authors evaluate the experimental results from a fair standpoint. Future research approaches with this study could be developed by combining reproducible experiments on a small laboratory scale, which can be done in a relatively short period of time, with chemical and biological evaluations after plasma treatment (this is just my comment).
Entire paper:
Names of microorganisms should be italicized.
Line 39:
"2" in "H2O2" would be expressed as a subscript.
Figure 1, 3, 6:
The position of legend in the figure could be adjusted a little more.
L104:
Since the seeds of the different hybrids must have been harvested on different farms, was the soil insect pest similarly infecting the different hybrid seeds? Or were they infecting the entire soil in which the experiment was conducted?
L107 “which were applied to maize seeds with the purpose of controlling soil pests”:
Fungicide and the eco-layer are usually applied to control not insect pests but microorganisms. How do you think about the effect of these? Is there any possibility of the combination effect with microorganisms and insects in 2020 results?
L190:
“2020” would be “2021”.
Author Response
Reviewer 1
Although this is a study in a realistic environment with various uncertainties, the value of this paper is in the fact that it has experimented with different types of treatments on various hybrid seeds and authors have found a certain degree of trend while being affected by pests. In this research field of plasma agriculture, there are few reports of research from this perspective, which requires a long period of time, and this research conducted over a two-year period is a suggestive contribution to this field. The disclosure of negative and positive data without selection is valuable because authors evaluate the experimental results from a fair standpoint. Future research approaches with this study could be developed by combining reproducible experiments on a small laboratory scale, which can be done in a relatively short period of time, with chemical and biological evaluations after plasma treatment (this is just my comment).
Entire paper:
- Names of microorganisms should be italicized.
We italicized all the names of microorganisms.
Line 39: "2" in "H2O2" would be expressed as a subscript.
We changed 2 into subscript.
- Figure 1, 3, 6: The position of legend in the figure could be adjusted a little more.
We corrected the positions of the legends.
- L104: Since the seeds of the different hybrids must have been harvested on different farms, was the soil insect pest similarly infecting the different hybrid seeds? Or were they infecting the entire soil in which the experiment was conducted?
The different hybrids were harvested on the same farm.
- L107 “which were applied to maize seeds with the purpose of controlling soil pests”: Fungicide and the eco-layer are usually applied to control not insect pests but microorganisms. How do you think about the effect of these? Is there any possibility of the combination effect with microorganisms and insects in 2020 results?
From Figure 1 (a) we cannot conclude that there is any combination effect with microorganisms and insects.
- L190: “2020” would be “2021”.
Thank you for pointing out our mistake, we changed the text.
Reviewer 2 Report
Review for Germination and Growth of Plasma-Treated Maize Seeds Planted in Fields and Exposed to Realistic Environmental Conditions by Nina Recek * , Rok Zaplotnik , Alenka Vesel , Gregor Primc , Peter Gselman , Miran Mozetic , Matej Holc.
Authors evaluate the effect of applying an inductively coupled, radio frequency oxygen plasma to maize seeds. They compared seedling emergency, plant number at tasseling, and crop yield of maize of several hybrids treated with plasma, fungicide, and an eco-layer. The study was performed over two years, however, during the first year the crop was affected by a plague which drastically reduced crop yield. In the second year, plasma application showed some positive effects on crop yield but were hybrid-dependent.
The manuscript in general is well written, but some details of methods are needed, for instance, the plant density or sowing density. Also, a better explanation of mechanisms by which plasma enhances germination or crop yield would be useful for readers.
Authors should consider eliminating data from the first crop season since this was severely affected by the insect pest. In another case, they should use some measures like the weight of 1000 grains or kernels and rows of ears for standardizing differences in the number of plants that were harvested.
Authors explain their results based on pests and climate variability. Thus, they should account for this effect in statistical models or at least showing precipitation, temperature, and solar radiation plots by crop season.
Finally, for this journal, I would expect a study focused on molecular/physiological mechanisms that cause better germination, emergence, crop yield, or any resistance to plagues. I recommend submitting this manuscript to another journal like “Agriculture”.
More comments into the attached document.
L80: which was the total number of seeds? germination percent is clearer.
L92: homogenize legends of plots
L137: There is no correlation at all.
L207: could you explain which correlation? There is no correlation in Fig. 6b.
Etc.

Author Response
Reviewer 2
Review for Germination and Growth of Plasma-Treated Maize Seeds Planted in Fields and Exposed to Realistic Environmental Conditions by Nina Recek *, Rok Zaplotnik, Alenka Vesel, Gregor Primc, Peter Gselman, Miran Mozetic, Matej Holc.
Authors evaluate the effect of applying an inductively coupled, radio frequency oxygen plasma to maize seeds. They compared seedling emergency, plant number at tasseling, and crop yield of maize of several hybrids treated with plasma, fungicide, and an eco-layer. The study was performed over two years, however, during the first year the crop was affected by a plague which drastically reduced crop yield. In the second year, plasma application showed some positive effects on crop yield but were hybrid-dependent.
The manuscript in general is well written, but some details of methods are needed, for instance, the plant density or sowing density. Also, a better explanation of mechanisms by which plasma enhances germination or crop yield would be useful for readers.
We thank the reviewer for the comment. We added the text about sowing density.
We agree that explaining mechanisms by which plasma enhances germination would be useful. However, in this paper, we mainly want to illustrate and show the growth of plasma-treated maize seeds planted in fields and exposed to realistic environmental conditions. There is a lack of such papers published and data obtained outside the controlled laboratory conditions. This is not a research paper studying the mechanisms of plasma on germination but rather showing the impact of plasma treatment in real conditions on a big scale (not a laboratory experiment).
Authors should consider eliminating data from the first crop season since this was severely affected by the insect pest. In another case, they should use some measures like the weight of 1000 grains or kernels and rows of ears for standardizing differences in the number of plants that were harvested.
Thank you for your valuable comment. We have considered your opinion; however, we feel it is important to keep the data from the first crop season, even if it was severely affected by the insect pest. This way, we considered all the variables in the real environment, which we believe is very important if we want to show accurate data. Suppose we show data for more consecutive years. In that case, the harvest can be severely infected by weather conditions (drought or floods), but this still would be important realistic data for this kind of study in realistic environmental conditions.
Authors explain their results based on pests and climate variability. Thus, they should account for this effect in statistical models or at least showing precipitation, temperature, and solar radiation plots by crop season.
Thank you for your comment. We agree that showing these effects in statistical models would be very useful. However, it is almost impossible to consider all the variables in the real environment and show them in statistical models. There are too many variables to consider, and impossible to gather all the climate data in a harvest year in one diagram.
However, we have explained the climate data (precipitation and temperature) in the comment to L212 below (see below).
Finally, for this journal, I would expect a study focused on molecular/physiological mechanisms that cause better germination, emergence, crop yield, or any resistance to plagues. I recommend submitting this manuscript to another journal like “Agriculture”.
Thank you. As already mentioned, this is not a paper in which we studied the different mechanisms of plasma on crops or on germination etc., but rather showing effects of plasma on crops in the real environment conditions on a big scale.
For this Journal, we got an invitation and recommendation to submit a paper to this special issue from the guest editor. This is why we believe it is a good choice for the Journal, although we did not study the mechanisms.
More comments are in the attached document:
L80: which was the total number of seeds? germination percent is clearer.
The total number of sowed seeds was 8.5 per m2; we added this value to the text.
L92: homogenize legends of plots
We homogenized legends on the plots.
L137: There is no correlation at all.
We apologize if we were not clear enough in the main text. We want to explain that there is a good correlation between the number of plants (seedlings) at emergence/m2 (Figure 3 a) and the number of grown plants later at tasseling time (Figure 3b). There is no significant difference in the number of emerged plants and grown plants at tasseling time, so this confirms there was no serious damage from insects or other pests.
L207: could you explain which correlation? There is no correlation in Fig. 6b.
There is a correlation that more tasselling plants per m2 produce more maize cobs, resulting in higher yield [kg/ha] and vice versa.
L208: Do you have "standard" data like the weight of 100 grains?
There is no standard data; it all depends on the size of corn kernels, so we should take average data, which is approximately 34 g for 100 seeds.
L212: Do you have precipitation and temperature data? It is valuable information for supporting this statement.
We have data about average day temperature and precipitation available for the whole year 2020 and 2021, for each day. This is a huge file of data, most importantly – in 2020, there were no extreme temperatures, but as expected during each of the four seasons. Considering the precipitation, there was lots of rain the summer season, more than usual – this was for sure one of the important factors considering the number of tasseling plants, yield etc in this harvest year. The year 2021 was “normal” considering the temperatures, as well as precipitation.
L283: specify "with plasma"
This sentence was misleading we changed it.
L293: More details of sowing are needed. Plant densities, size of blocks, etc.
Seeding density was 8,5 per m2. The size of block was 19.6 m2 (length 7 m and width 2.8 m). We added this information in the text.
L321: Unnecessary
Thank you for your comment. We feel some valuable information is included, so we have kept this part.
L328: This is not a consequence of treatments.
Correct, not the treatment but the insect Agriotes spp., as we explained in the text (this is probably what you meant with your comment).
L329: This looks more like results; consider combining this into the next paragraph
Thank you for your comment. We agree and it was well metioned and explained in more detail in the result section.
L336: Could you consider this "random" effect into the statistical model?
Unfortunately, we can not consider these effects in the statistical model: as we explained, there are too many variations, from pests' attacks to field growth and weather conditions.
Round 2
Reviewer 2 Report
Authors addressed the majority of comments; however. There are some issues that should be clarified/correct before the manuscript could be accepted.
The manuscript in general is well written, but some details of methods are needed, for instance, the plant density or sowing density. Also, a better explanation of mechanisms by which plasma enhances germination or crop yield would be useful for readers.
We thank the reviewer for the comment. We added the text about sowing density.
OK
We agree that explaining mechanisms by which plasma enhances germination would be useful. However, in this paper, we mainly want to illustrate and show the growth of plasma-treated maize seeds planted in fields and exposed to realistic environmental conditions. There is a lack of such papers published and data obtained outside the controlled laboratory conditions. This is not a research paper studying the mechanisms of plasma on germination but rather showing the impact of plasma treatment in real conditions on a big scale (not a laboratory experiment).
OK, but should be very informative what to expect about plasma treatment on crop yield. If you only expect effects of treatments on germination, enhancement of crop yield would be a result of the largest number of plants at harvest? This will lead to establishing one hypothesis.
Maybe explain “why” or “how” in lines 38 - 40
Authors should consider eliminating data from the first crop season since this was severely affected by the insect pest. In another case, they should use some measures like the weight of 1000 grains or kernels and rows of ears for standardizing differences in the number of plants that were harvested.
Thank you for your valuable comment. We have considered your opinion; however, we feel it is important to keep the data from the first crop season, even if it was severely affected by the insect pest. This way, we considered all the variables in the real environment, which we believe is very important if we want to show accurate data. Suppose we show data for more consecutive years. In that case, the harvest can be severely infected by weather conditions (drought or floods), but this still would be important realistic data for this kind of study in realistic environmental conditions.
OK
Authors explain their results based on pests and climate variability. Thus, they should account for this effect in statistical models or at least showing precipitation, temperature, and solar radiation plots by crop season.
Thank you for your comment. We agree that showing these effects in statistical models would be very useful. However, it is almost impossible to consider all the variables in the real environment and show them in statistical models. There are too many variables to consider, and impossible to gather all the climate data in a harvest year in one diagram.
However, we have explained the climate data (precipitation and temperature) in the comment to L212 below (see below).
This needs more effort, but if you include a plot with precipitation and temperature will be helpful -see bellow-.
Finally, for this journal, I would expect a study focused on molecular/physiological mechanisms that cause better germination, emergence, crop yield, or any resistance to plagues. I recommend submitting this manuscript to another journal like “Agriculture”.
Thank you. As already mentioned, this is not a paper in which we studied the different mechanisms of plasma on crops or on germination etc., but rather showing effects of plasma on crops in the real environment conditions on a big scale.
For this Journal, we got an invitation and recommendation to submit a paper to this special issue from the guest editor. This is why we believe it is a good choice for the Journal, although we did not study the mechanisms.
OK
More comments are in the attached document:
L80: which was the total number of seeds? germination percent is clearer.
The total number of sowed seeds was 8.5 per m2; we added this value to the text.
OK
L92: homogenize legends of plots
We homogenized legends on the plots.
OK
L137: There is no correlation at all.
We apologize if we were not clear enough in the main text. We want to explain that there is a good correlation between the number of plants (seedlings) at emergence/m2 (Figure 3 a) and the number of grown plants later at tasseling time (Figure 3b). There is no significant difference in the number of emerged plants and grown plants at tasseling time, so this confirms there was no serious damage from insects or other pests.
OK
L207: could you explain which correlation? There is no correlation in Fig. 6b.
There is a correlation that more tasselling plants per m2 produce more maize cobs, resulting in higher yield [kg/ha] and vice versa.
This is clear in Fig. 2, but not in all hybrids. Which treatment is plotted? I understand different colors stand for different hybrids, but it does not tell us which treatment. Are all treatments pooled into the different hybrids?
In Fig5b and 6b data are not correlated, maybe this is because of the short range of plants (n<1), in contrast with Fig. 2. In fact, a variation in the x-axis of less than one plant doesn’t make sense. You should test with and LS or Theil-Sen regression analysis whether there is any significant correlation between number of plants at tasseling and yield. Change the sentence in L206-208 accordingly.
L208: Do you have "standard" data like the weight of 100 grains?
There is no standard data; it all depends on the size of corn kernels, so we should take average data, which is approximately 34 g for 100 seeds.
OK
L212: Do you have precipitation and temperature data? It is valuable information for supporting this statement.
We have data about average day temperature and precipitation available for the whole year 2020 and 2021, for each day. This is a huge file of data, most importantly – in 2020, there were no extreme temperatures, but as expected during each of the four seasons. Considering the precipitation, there was lots of rain the summer season, more than usual – this was for sure one of the important factors considering the number of tasseling plants, yield etc in this harvest year. The year 2021 was “normal” considering the temperatures, as well as precipitation.
You could summarize both monthly precipitation and temperature in one plot. This is a valuable information for understanding differences in crop yield between years (apart of insects). You also could mention average annual precipitation and mean temperature. Please, provide a plot; it is simple, but very informative.
L283: specify "with plasma"
This sentence was misleading we changed it.
OK
L293: More details of sowing are needed. Plant densities, size of blocks, etc.
Seeding density was 8,5 per m2. The size of block was 19.6 m2 (length 7 m and width 2.8 m). We added this information in the text.
OK
L321: Unnecessary
Thank you for your comment. We feel some valuable information is included, so we have kept this part.
OK
L328: This is not a consequence of treatments.
Correct, not the treatment but the insect Agriotes spp., as we explained in the text (this is probably what you meant with your comment).
OK
L329: This looks more like results; consider combining this into the next paragraph
Thank you for your comment. We agree and it was well metioned and explained in more detail in the result section.
OK
L336: Could you consider this "random" effect into the statistical model?
Unfortunately, we can not consider these effects in the statistical model: as we explained, there are too many variations, from pests' attacks to field growth and weather conditions.
OK
Author Response
Reply to the Reviewer #2
We agree that explaining mechanisms by which plasma enhances germination would be useful. However, in this paper, we mainly want to illustrate and show the growth of plasma-treated maize seeds planted in fields and exposed to realistic environmental conditions. There is a lack of such papers published and data obtained outside the controlled laboratory conditions. This is not a research paper studying the mechanisms of plasma on germination but rather showing the impact of plasma treatment in real conditions on a big scale (not a laboratory experiment).
OK, but should be very informative what to expect about plasma treatment on crop yield. If you only expect effects of treatments on germination, enhancement of crop yield would be a result of the largest number of plants at harvest? This will lead to establishing one hypothesis.
Maybe explain “why” or “how” in lines 38 – 40.
We explained our hypothesis in the Introduction section; see lines 40-45.
Thank you for your comment. We agree that showing these effects in statistical models would be very useful. However, it is almost impossible to consider all the variables in the real environment and show them in statistical models. There are too many variables to consider, and impossible to gather all the climate data in a harvest year in one diagram.
However, we have explained the climate data (precipitation and temperature) in the comment to L212 below (see below).
This needs more effort, but if you include a plot with precipitation and temperature will be helpful -see bellow-.
We added a figure with precipitation and temperature plot (see below).
L212: Do you have precipitation and temperature data? It is valuable information for supporting this statement.
We have data about average day temperature and precipitation available for the whole year 2020 and 2021, for each day. This is a huge file of data, most importantly – in 2020, there were no extreme temperatures, but as expected during each of the four seasons. Considering the precipitation, there was lots of rain the summer season, more than usual – this was for sure one of the important factors considering the number of tasseling plants, yield etc in this harvest year. The year 2021 was “normal” considering the temperatures, as well as precipitation.
You could summarize both monthly precipitation and temperature in one plot. This is a valuable information for understanding differences in crop yield between years (apart of insects). You also could mention average annual precipitation and mean temperature. Please, provide a plot; it is simple, but very informative.
We inserted a plot of Average daily temperature and Precipitation for years 2020 and 2021 at the end of “Results” section and added some discussion about these data.
There is a correlation that more tasselling plants per m2 produce more maize cobs, resulting in higher yield [kg/ha] and vice versa.
This is clear in Fig. 2, but not in all hybrids. Which treatment is plotted? I understand different colors stand for different hybrids, but it does not tell us which treatment. Are all treatments pooled into the different hybrids?
In Fig5b and 6b data are not correlated, maybe this is because of the short range of plants (n<1), in contrast with Fig. 2. In fact, a variation in the x-axis of less than one plant doesn’t make sense. You should test with and LS or Theil-Sen regression analysis whether there is any significant correlation between number of plants at tasseling and yield. Change the sentence in L206-208 accordingly.
The reviewer is correct, thanks for pointing this out. We did a test with LS and indeed there were no correlation also in Figure 6b. Therefore, we changed the sentence accordingly.
Round 3
Reviewer 2 Report
All my suggestions were addressed correctly, but one of them is unresolved,
In fig. 2, 4, 5b and 6b, which treatment is plotted? I understand different colors stand for different hybrids, but it does not tell us which treatment (Control, Plasma, fungicide, etc.). Are all treatments pooled into the different hybrids?
Please clarify the treatments in those figures.
Finally, precipitation is a cumulative variable; thus, it is better to plot it like bars. Please modify Fig. 7; bars (columns) for precipitation and lines for temperature.
The manuscript should be publishable after these minor changes.
Author Response
Reply to the Reviewer #2
All my suggestions were addressed correctly, but one of them is unresolved.
In fig. 2, 4, 5b and 6b, which treatment is plotted? I understand different colors stand for different hybrids, but it does not tell us which treatment (Control, Plasma, fungicide, etc.). Are all treatments pooled into the different hybrids?
Please clarify the treatments in those figures.
We are sorry, we missed this one. Yes, all treatments are pooled into different hybrids, therefore 5 points (4 treatments + 1 control) of the same color (same hybrid) are presented. We also explained this in the manuscript.
Finally, precipitation is a cumulative variable; thus, it is better to plot it like bars. Please modify Fig. 7; bars (columns) for precipitation and lines for temperature.
We changed the plot in Figure 7 as suggested by the reviewer.